# Modulating Nitric Oxide: Implications for Cytotoxicity and Cytoprotection

**DOI:** 10.3390/antiox13050504

**Published:** 2024-04-23

**Authors:** Igor Belenichev, Olena Popazova, Nina Bukhtiyarova, Dmytro Savchenko, Valentyn Oksenych, Oleksandr Kamyshnyi

**Affiliations:** 1Department of Pharmacology and Medical Formulation with Course of Normal Physiology, Zaporizhzhia State Medical and Pharmaceutical University, 69000 Zaporizhzhia, Ukraine; 2Department of Histology, Cytology and Embryology, Zaporizhzhia State Medical and Pharmaceutical University, 69000 Zaporizhzhia, Ukraine; 3Department of Clinical Laboratory Diagnostics, Zaporizhzhia State Medical and Pharmaceutical University, 69000 Zaporizhzhia, Ukraine; 4Department of Pharmacy and Industrial Drug Technology, Bogomolets National Medical University, 01601 Kyiv, Ukraine; 5Broegelmann Research Laboratory, Department of Clinical Science, University of Bergen, 5020 Bergen, Norway; 6Department of Microbiology, Virology and Immunology, I. Horbachevsky Ternopil State Medical University, 46001 Ternopil, Ukraine; kamyshnyi_om@tdmu.edu.ua

**Keywords:** nitric oxide, inducible nitric oxide synthases, iNOS, endothelial nitric oxide synthases, eNOS, heat shock proteins, HSP

## Abstract

Despite the significant progress in the fields of biology, physiology, molecular medicine, and pharmacology; the designation of the properties of nitrogen monoxide in the regulation of life-supporting functions of the organism; and numerous works devoted to this molecule, there are still many open questions in this field. It is widely accepted that nitric oxide (^•^NO) is a unique molecule that, despite its extremely simple structure, has a wide range of functions in the body, including the cardiovascular system, the central nervous system (CNS), reproduction, the endocrine system, respiration, digestion, etc. Here, we systematize the properties of ^•^NO, contributing in conditions of physiological norms, as well as in various pathological processes, to the mechanisms of cytoprotection and cytodestruction. Current experimental and clinical studies are contradictory in describing the role of ^•^NO in the pathogenesis of many diseases of the cardiovascular system and CNS. We describe the mechanisms of cytoprotective action of ^•^NO associated with the regulation of the expression of antiapoptotic and chaperone proteins and the regulation of mitochondrial function. The most prominent mechanisms of cytodestruction—the initiation of nitrosative and oxidative stresses, the production of reactive oxygen and nitrogen species, and participation in apoptosis and mitosis. The role of ^•^NO in the formation of endothelial and mitochondrial dysfunction is also considered. Moreover, we focus on the various ways of pharmacological modulation in the nitroxidergic system that allow for a decrease in the cytodestructive mechanisms of ^•^NO and increase cytoprotective ones.

## 1. Introduction

**Nitric oxide.** Nitric oxide (^•^NO) is a unique molecule that, despite its extremely simple structure, plays a pivotal role in various physiological processes within the body. Its multifunctionality has spurred intensive research over the past decade [1,2,3]. ^•^NO serves as a crucial element in the cardiovascular system, facilitating vasodilation and regulating blood pressure. Additionally, it participates in signaling within both the central and peripheral nervous systems [4,5,6]. 

It has been demonstrated that ^•^NO is essential for exerting cytotoxic effects on tumor cells and cells affected by viruses. In this context, the mechanism of action of nitric oxide does not involve the activation of guanylate cyclase but is primarily attributed to the direct effects of ^•^NO itself. It is noteworthy that the free-radical nature of the nitric oxide molecule, characterized by the presence of an unpaired electron at the nitrogen atom, renders it highly reactive. In vivo, ^•^NO has an average lifetime of 5–30 s, during which it rapidly interacts with its targets, primarily thiols and transition metals, or undergoes oxidation to form inactive nitrate and nitrite, for instance, via cytochrome C oxidase [7,8], or ^•^NO may generate reactive oxygen species. Hence, the action of ^•^NO can occur through direct or indirect mechanisms. Direct effects result from the reactions of ^•^NO itself with its targets, such as the stimulation of guanylate cyclase or the formation of nitrosyl complexes with metals, often leading to the inactivation of enzymes containing these metal ions. The indirect effects of ^•^NO are defined as chemical reactions mediated by active forms of nitric oxide, which are generated through interactions with superoxide (O^2−^) or oxygen (O^2^). The action of these active forms of ^•^NO leads to the development of nitrosylation stress (resulting in the formation of nitrosoamines, S-nitrosothiols, and the deamination of DNA bases) or oxidative stress [9]. Owing to its high lipophilicity, ^•^NO efficiently penetrates membranes, enabling it to diffuse from its source to distances several times the size of the cell, thus affecting its targets within this extended range [10,11].

## 2. Basic Mechanisms of Nitric Oxide (^•^NO) Regulation

### 2.1. ^•^NO Synthesis in the Body

This investigation into the origin of endogenous ^•^NO has revealed that L-arginine is essential for its production by active macrophages. Subsequently, it was discovered that a family of enzymes called nitric oxide synthases (NOS) is responsible for ^•^NO production. These enzymes catalyze the conversion of L-arginine into ^•^NO and L-citrulline, simultaneously utilizing NADPH and reducing oxygen to water [12,13,14]. NOS enzymes are ubiquitously present in the cells of almost all tissue types and are categorized into constitutive (cNOS) and inducible (iNOS) forms based on their expression patterns. The cNOS group typically includes neuronal (ncNOS or NOS1) and endothelial (ecNOS or NOS3) isoforms, with primary localization in neurons and endothelial cells, respectively, although they are also found in other cell types. iNOS is predominantly associated with macrophages and plays a key role in the immune system [15,16,17,18,19,20]. Its expression increases in response to activation by cytokines (such as IFN-g, IL-1b, and TNF-a) and other agents like lipopolysaccharides (LPS). This isoform is also expressed in the liver upon stimulation, which is associated with the barrier function of this organ [21,22,23]. A comprehensive study of NOS has revealed that they are among the most intricately structured and regulated enzymes, boasting an unusually high number of cofactors. NOS exist in the cell as dimers and are active only in this state. Within each subunit of the dimer, distinct domains such as reductase, calmodulin-binding, and oxygenase can be identified. The reductase domain contains the flavins FAD and FMN: FAD serves as the primary electron acceptor from NADPH, while FMN transfers electrons from FAD to the heme of the oxygenase domain. The oxygenase domain contains heme, arginine (L-Arg), and tetrahydrobiopterin (BH4) binding sites. Calmodulin-Ca^2+^ is believed to confer the enzyme with the necessary conformation for internal electron transfer [24,25,26,27]. It is the variations in the binding affinity of calmodulin to the NOS dimer that underlie the catalytic discrepancies between the isoforms: the activity of nNOS and eNOS is highly reliant on Ca^2+^ concentration, whereas calmodulin binds to iNOS so tightly that Ca^2+^ supplementation is unnecessary. Although the specific activities of all NOS isoforms are comparable, in vivo, it seems that cNOS produces small amounts of ^•^NO over brief intervals, while iNOS generates much larger quantities of ^•^NO over extended periods (up to several days). Thus, the expression and activity of a specific isoform may dictate ^•^NO’s role as either a physiological modulator or a cytotoxic agent [28,29,30,31]. 

In vitro studies of ^•^NO-mediated macrophage cytotoxicity have unequivocally demonstrated that the addition of NOS inhibitors, such as the substrate analogue NG-monomethyl-L-arginine (L-NMMA), to the medium suppresses the cytotoxic effect of macrophages on tumour cells. This supports the prevailing role of ^•^NO in mediating the macrophage’s effect on target cells. However, it is important to acknowledge the well-known phenomenon of the respiratory burst, which also plays a crucial role in pathogen destruction by phagocytes. Additionally, recent data have emerged that complicate the understanding of macrophage cytotoxicity induced by nitric oxide. Specifically, it has been discovered that wound macrophages capable of ^•^NO production do not exhibit cytotoxicity towards ^•^NO-sensitive cells of the P815 line. Thus, the question arises regarding the necessity and sufficiency of ^•^NO for the cytotoxicity of macrophages [32,33,34,35,36]. It should also be noted that ^•^NO production can have significant negative effects on the macrophages that produce it. Studies have demonstrated that phagocytosis and the production of reactive oxygen species are markedly inhibited in rat or peritoneal macrophages cultured under conditions that allow ^•^NO production. Macrophages that express iNOS or are treated with nitric oxide exhibit nuclear and cytoplasmic condensation. Therefore, the release of ^•^NO by activated macrophages leads to their functional suppression, eventually resulting in apoptosis. These phenomena are clearly attributed to ^•^NO, as they can be prevented by the addition of NOS inhibitors [37,38].

### 2.2. Mechanisms of ^•^NO Cytotoxicity

Nitric oxide targets are currently under active investigation to determine whether ^•^NO itself is sufficiently cytotoxic or if its derivatives are more potent [39]. ^•^NO in target cells is known to generate active intermediates such as nitrosonium (NO^+^), nitroxyl (NO^−^), and peroxynitrite (ONOO^−^). Some researchers posit that most of the cytotoxic effects attributed to ^•^NO actually stem from ONOO^−^, which forms through reaction with superoxide (O^2−^). Indeed, peroxynitrite is significantly more reactive; it extensively nitrosylates proteins and can serve as a source of highly toxic hydroxyl radicals (-OH) through reactions [39,40,41,42].
^•^NO · + O^2−^ a ONOO^−^ + H^+^ a ONOOH a ONO + · OH

OH causes lipid peroxidation and other phenomena associated with oxidative stress. Another challenge encountered in this study of the mechanisms of nitrogen cytotoxicity is related to the ^•^NO donors utilized for its generation, as described above. Specifically, S-nitrosothiols (mainly GSNO and SNAP), frequently employed in many studies, have the capability to engage in transnitrosylation reactions. These reactions involve the transfer of the NO^+^ group to thiols (such as glutathione and the SH-groups of proteins), thereby disrupting their cellular functions [43,44,45,46]. However, it remains unclear whether such reactions should be attributed solely to the effects of ^•^NO itself. Some possible mechanisms of the cytotoxic action of nitric oxide will be discussed below, and some reactions can be induced by its derivatives. It is demonstrated that ^•^NO (from macrophages or exogenously administered) primarily inhibits oxidative phosphorylation in the mitochondria of target cells [47,48]. This inhibition occurs because ^•^NO reversibly binds to cytochrome-C-oxidase of the mitochondrial electron transport chain. However, the inhibition of electron transport in the mitochondrion leads to the generation of superoxide and subsequently the formation of peroxynitrite. The conversion of nitric oxide to peroxynitrite involves a reaction between two radicals: O^2−^ and ^•^NO, resulting in the formation of ONOO^−^, a potent oxidant in the mitochondrial matrix. Normally, ONOO is reduced by mitochondrial reducing agents such as NADH2, ubiquinol UQH2, and glutathione GSH. However, when produced in excess due to loss of control (e.g., during ischemia/reperfusion or inflammation), it leads to tyrosine nitration and mitochondrial dysfunction. Its cumulative effect contributes to tissue aging. Another radical widely produced by both enzymatic and non-enzymatic processes is ^•^NO, which serves as an intra- and intercellular signaling molecule [49,50]. ^•^NO and superoxide react in a diffusion-limited manner. This reaction halts the chain reaction initiated by superoxide, although peroxynitrite is generally considered a harmful molecule [51]. Peroxynitrite is a short-lived and highly reactive oxidant, thus representing another mechanism that imparts indirect toxicity to O^2−^, particularly targeting DNA, proteins, and lipids. Additionally, peroxynitrite has the capability to nitrate tyrosine or tryptophan residues or oxidize methionine residues [52,53]. This suppression of mitochondrial respiration leads to a decrease in the mitochondrial membrane potential, which can initiate the apoptotic process [54]. Conversely, it is known that in the absence of glucose or when glycolysis is blocked, the ^•^NO-induced suppression of respiration leads to necrosis rather than apoptosis [55]. There is also evidence of the direct activation of giant pore opening by ^•^NO, leading to the release of cytochrome C and the initiation of the caspase cascade. However, this is also controversial, as other investigators have shown that low doses of ^•^NO (close to physiological doses) slow giant pore opening and apoptosis, whereas peroxynitrite (ONOO^−^) and nitrosothiols promote them. ^•^NO and its derivatives can cause the peroxidation of phospholipids and the oxidation of thiol groups of mitochondrial membrane proteins, which also lead to the release of apoptogenic factors into the cytosol [56,57,58]. 

The nitrosylation of proteins at tyrosine residues by peroxynitrite (ONOO^−^) can have profound functional consequences, as it inhibits Tyr phosphorylation, thus disrupting vital signal transduction pathways within the cell. Recent studies have revealed that peroxynitrite can also nitrosylate cytochrome C within mitochondria, altering its function significantly. This modification renders cytochrome C incapable of supporting electron transport in the respiratory chain and is resistant to reduction by ascorbate. Concurrently, nitrated cytochrome C is released into the cytoplasm, suggesting potential involvement in signaling processes [59,60]. Emerging hypotheses propose that selective protein nitrosylation may function as a regulatory mechanism akin to phosphorylation [61,62]. Peroxynitrite’s activity extends beyond protein modification to include guanine nitrosylation and DNA strand breaks, which can instigate mutations or trigger apoptosis pathways [63]. Additionally, ^•^NO exerts effects on DNA repair enzymes by inhibiting their activity. Notably, different ^•^NO donors impact various enzymes responsible for DNA repair, such as alkyltransferase, formamidopyrimidine-DNA-glycosylase, and ligase, suggesting a multifaceted role for ^•^NO in genomic integrity maintenance and cellular signaling processes [9,64]. 

It is well-established that ^•^NO can activate Poly(ADP-ribose) polymerase (PARP) and induce ADP-ribosylation, potentially as a response to DNA damage. However, this activation tends to lead to necrosis due to the depletion of the NAD and ATP pools, rather than triggering apoptosis. Regarding ^•^NO and its derivatives’ impact on DNA, investigations into their effect on p53 expression are particularly intriguing. p53 is a crucial protein involved in tumor suppression, genome maintenance, and the regulation of cell cycle progression or apoptosis. It is known that p53 can upregulate the expression of pro-apoptotic proteins such as Bax, Fas, and p53AIP (apoptosis-inducing protein), and other apoptogenic proteins [65,66,67,68]. Additionally, during apoptosis, p53 translocates into the mitochondrion, which may be one of the reasons for the production of ROS [69,70].

Under normal conditions, the cellular concentration of p53 remains low as it undergoes rapid degradation. However, DNA damage triggers the accumulation of p53. Experiments conducted on macrophages and RINm5F insulinoma cells have demonstrated p53 accumulation in ^•^NO-induced cell death scenarios [71,72]. Further research revealed that L-NMMA, a nitric oxide synthase (NOS) inhibitor, suppresses p53 accumulation induced by cytokines or lipopolysaccharides (LPS), indicating an active involvement of ^•^NO in this process. Some evidence suggests that ^•^NO’s effect on p53 accumulation may be linked to its ability to inhibit proteasome function, thus interfering with p53 degradation pathways [73,74]. Nevertheless, further experiments have uncovered the operation of p53-independent pathways in ^•^NO-induced apoptosis [75,76]. Additional studies have elucidated a negative feedback mechanism between the levels of ^•^NO and p53 in various human cell types: the accumulation of ^•^NO leading to DNA damage triggers the expression of p53, which in turn suppresses the human inducible nitric oxide synthase (iNOS) gene [77]. Additionally, ^•^NO represses iNOS expression by dampening NFκB activity in hepatocytes. Through these intricate pathways, the precise regulation of ^•^NO synthesis is achieved, thus mitigating its deleterious effects on the tissue (Figure 1) [78,79,80]. 

### 2.3. Involvement of ^•^NO in the Formation of Mitochondrial Dysfunction and Mitoptosis 

Since the involvement of mitochondria and ^•^NO in apoptosis has been extensively elucidated in this study, it is pertinent to describe their combined role in its regulation. In experiments involving the transfection of the RAW264.7 macrophage line with human Bcl-2, the transfected cells exhibited protection against death induced by inducible nitric oxide synthase (iNOS) activation [81,82,83,84]. This suggests that Bcl-2 operates by nullifying the ^•^NO-induced upregulation of Bax protein expression. Additional experiments demonstrated that P815 tumor line cells transfected with Bcl-2 showed resistance to the effects of the ^•^NO donor SNAP (S-nitroso-N-acetylpenicillin-amine) and to ^•^NO-associated cytotoxicity from activated murine macrophages [85,86]. Furthermore, L929 cells overexpressing Bcl-2 were shielded from apoptosis triggered by iNOS activation. A multitude of other instances showcasing the interaction between ^•^NO and Bcl-2 are provided in articles [87,88,89,90,91]. 

The interaction of ^•^NO with members of the Bcl-2 superfamily is further evidenced by the significant reduction in intracellular Bcl-2 protein levels upon exposure to ^•^NO within the cell [92]. This reduction may occur through caspase-induced cleavage or p53-dependent suppression of its expression, although conflicting evidence exists regarding this mechanism [93,94]. Additionally, the proapoptotic effect of nitric oxide manifests through its inducible increase in Bax expression [95]. Apart from the aforementioned mitochondrial functions, recent studies have shed light on the involvement of mitochondria not only in the reception of apoptotic signals from ^•^NO but also in the production of ^•^NO itself. Notably, a constitutive form of nitric oxide synthase (NOS) has been identified within mitochondria [96,97]. 

The initial detection of ^•^NO production in rat liver mitochondria prompted further investigation into the purification of mitochondrial NOS and the elucidation of its enzymatic characteristics [98]. This isoform of NOS appears to be localized within the mitochondrial membrane, particularly in the inner membrane [99]. It has been revealed that mitochondrial nitric oxide synthase (mtNOS) bears a striking resemblance to macrophage inducible NOS (iNOS) but is expressed constitutively. However, the classification of mtNOS as a distinct isoform remains uncertain, as it remains unclear whether it represents a separate isoform or if it is iNOS undergoing post-translational modifications that dictate its distinct subcellular localization. Notably, mtNOS exhibits independence from calmodulin and calcium addition, indicating a strong association with calmodulin. Purified mtNOS, when subjected to suboptimal concentrations of L-Arginine, demonstrates a capability to generate O^2−^, albeit at a relatively modest rate. This observation aligns with the documented homology of the C-terminal domain of NOS to NADPH: cytochrome P450-oxidoreductase, which also possesses NADPH-oxidase activity and generates O^2−^, albeit at a rate approximately tenfold faster than mtNOS [100,101,102,103,104,105,106,107]. 

The identification of such NOS within the mitochondrion raises numerous inquiries and suggests promising avenues for further investigation. Foremost among these is the inquiry into how the ^•^NO produced within mitochondria influences apoptosis. Considering the established role of non-mitochondrial ^•^NO, which acts directly on mitochondria, inducing a range of phenomena culminating in apoptosis, it is reasonable to speculate on the involvement of mtNOS in the regulation of apoptosis, although conclusive evidence is yet to be established [108,109]. Moreover, given its capacity to generate not only ^•^NO but also O^2−^, it may be implicated in the production of reactive oxygen species (ROS), potentially contributing to various biological damages. In the context of apoptosis, intriguing insights have emerged from investigations into the release of cytochrome C from mitochondria upon mtNOS stimulation [110,111]. While elevated cytosolic Ca^2+^ levels have long been recognized as apoptosis inducers, recent findings have underscored the significant role of mtNOS in this process. It has been demonstrated that this form of apoptosis necessitates mitochondrial Ca^2+^ uptake, triggering mtNOS activation and subsequent cytochrome C release into the cytosol. Concurrently, there is an augmentation in lipid peroxidation (LPO). Notably, the release of cytochrome C and LPO are effectively inhibited by NOS inhibitors (such as L-NMMA), peroxynitrite scavengers (e.g., urate), and Bcl-2 expression. These findings suggest that upon Ca^2+^-induced activation of mtNOS, peroxynitrite is generated within the mitochondria, leading to LPO and cytochrome C release, ultimately leading to a pattern of typical apoptosis. Further elucidation of these mechanisms promises to enhance our comprehension of the roles played by mitochondria and ^•^NO in various pathways of cell death [112,113,114,115,116]. 

For instance, recent findings have demonstrated that inhibition of mitochondrial nitric oxide synthase (mtNOS) results in the accumulation of intramitochondrial Ca^2+^, indicating that ^•^NO produced by mtNOS impedes Ca^2+^ accumulation. Given that the elevation in matrix Ca^2+^ concentration is responsible for altering mitochondrial membrane permeability, it is deduced (contrary to the earlier assertion) that mitochondrial ^•^NO decelerates the opening of the mitochondrial permeability transition pore and the subsequent release of cytochrome C [117,118,119,120,121,122,123]. While there are indications that the synthesis of ^•^NO may be regulated through substrates of mtNOS (such as L-Arginine, NADPH) and its cofactors (including FMN, FAD, BH4), akin to other NOS isoforms, this aspect remains largely unexplored. Furthermore, this endogenous mitochondrial ^•^NO may play a crucial role in regulating mitochondrial activity by inhibiting cytochrome oxidase (complex IV), as well as complexes I and II of the electron transport chain. Its reaction with oxygen could modulate mitochondrial respiration by altering the availability of oxygen for electron acceptance, thereby impacting cellular energy supply. Nevertheless, this facet of mitochondrial ^•^NO biology necessitates further investigation [124,125,126,127,128,129,130]. 

Summarizing all these data, we can say that the mitochondria is the central link where many signaling pathways of apoptosis converge and are regulated. Within these pathways, mitochondria may assume a central role by initiating the pro-apoptotic cascade, particularly under stressors such as irradiation or ^•^NO action. Conversely, mitochondria can also amplify certain apoptogenic signaling cascades, such as those mediated by the Fas receptor or TNF receptor, through kinase-mediated mechanisms. Besides superoxide, both ^•^NO and its more aggressive derivative, peroxynitrite, are instrumental in the genesis of mitochondrial abnormalities and apoptosis [131,132,133,134,135,136]. Notably, neuronal mitochondria emerge as a significant source of ^•^NO, with evidence demonstrating the presence of a constitutive form of NOS localized in the inner mitochondrial membrane and ^•^NO production within the mitochondria of hippocampal neurons. Moreover, mitochondrial NOS is capable of generating superoxide at suboptimal concentrations of L-arginine. Importantly, mitochondrial NOS exhibits significant activation in response to the onset of glutamate excitotoxicity and mitochondrial calcium uptake. Additionally, cytokines such as IL-1β and TNF-α contribute to the activation of mitochondrial NOS [137,138,139,140,141,142].

This leads to the generation of peroxynitrite, which facilitates the opening of the mitochondrial permeability transition pore (mPTP). Additionally, peroxynitrite nitrosylates cytochrome C within mitochondria, inducing alterations in its functionality. Specifically, this modification renders cytochrome C incapable of supporting electron transfer within the respiratory chain and is resistant to reduction by ascorbate. Concurrently, there is a concomitant release of cytochrome C, including nitrated forms, into the cytoplasm, suggesting the involvement of this nitrosylation process in various signaling pathways. Furthermore, peroxynitrite nitrosylates guanine, resulting in DNA strand breaks, mutations, or the activation of apoptosis-related processes [138,143,144,145]. 

Excessive ^•^NO inhibits enzymes crucial for DNA repair, targeting alkyltransferase, formamidopyrimidine-DNA glycosylase, and ligase. Moreover, ^•^NO activates PARP and ADP-ribosylation, particularly under conditions of ATP depletion and the accumulation of reduced pyridine nucleotides. Additionally, ^•^NO exerts a positive influence on the synthesis of the tumor suppressor protein p53. Enhanced p53 expression promotes the upregulation of pro-apoptotic proteins such as Bax, Fas, and p53AIP (apoptosis-inducing protein). Furthermore, ^•^NO translocates into the mitochondria during apoptosis, potentially contributing to the production of ROS and the reduction of transmembrane potential across the inner mitochondrial membrane [72,146,147,148,149,150]. Currently, there exists a widely recognized concept known as “mitochondrial dysfunction.” This represents a characteristic pathological process lacking etiological and nosological specificity. 

The progression of mitochondrial dysfunction precipitates the disturbance of mediator reuptake (e.g., catecholamines, dopamine, serotonin), ion transport, impulse generation and conduction, and de novo protein synthesis, as well as translation and transcription processes. Concurrently, “parasitic” energy-producing reactions are activated, resulting in a substantial depletion of neuronal cell energy reserves. Furthermore, the action of the hydroxyl radical triggers the opening of mitochondrial pores, leading to the expression and release of proapoptotic proteins into the cytosol. This pore opening occurs via the oxidation of thiol groups within the cysteine-dependent region of the mitochondrial inner membrane protein (specifically, the ATP/ADP antiporter), transforming it into a permeable nonspecific channel pore [151,152]. The opening of these pores transforms mitochondria from being “power plants” into “furnaces” for oxidation substrates, devoid of ATP production. Precise biochemical investigations have revealed that disturbances in tissue oxygenation, the hyperproduction of excitotoxic amino acids, decreased “normal” calcium (Ca^2+^) accumulation by mitochondria, and damage to mitochondrial membrane ROS all contribute to the increased opening of these pores and the subsequent release of apoptogenic proteins from damaged mitochondria. In this context, the pivotal role of the neurotrophic factor tumor necrosis factor-alpha (TNF-α) cannot be overstated, as it is intricately associated with the opening of mitochondrial pores, subsequent membrane disruption, and the onset of mitoptosis [138,153,154,155,156,157,158,159].

## 3. Effects of ^•^NO

### 3.1. Apoptosis and ^•^NO

Cu has been found to be repressed in neurons with evidence of apoptosis, while Zn-SOD is known to bind and facilitate the production of significant amounts of endogenous peroxynitrite (ONOO^−^) [160,161,162]. Low concentrations of ^•^NO donors have been shown to inhibit neuroapoptosis induced by growth factor deprivation or TNF-α addition, potentially through the activation of the heat shock protein HSP70 or the inhibition of caspase-3 [163,164]. Dinitrosyl iron complexes (DNICs) at concentrations ranging from 5 to 10 μM are believed to suppress IL-1β-induced neuroapoptosis by promoting ^•^NO synthesis. Conversely, elevating DNIC concentration (0.5 mM) has been associated with the induction of neuroapoptosis through the generation of ONOO^−^ [165,166,167,168,169].

Conditions that lead to an increased bioavailability of ^•^NO and a reduced formation of ONOO-, such as enhanced superoxide dismutase (SOD) activity and the presence of reduced thiol antioxidants, render neurons resistant to Fas-induced apoptosis. Conversely, the decreased bioavailability of ^•^NO and increased levels of cytotoxic forms of ^•^NO heighten cellular sensitivity to signals mediated through Fas receptors. Additionally, neuroapoptosis can be induced by the synergistic action of H_2_O_2_ and Fe^2+^, which convert ^•^ to peroxynitrite, while the depression of bcl-2 and induction of c-fos were simultaneously observed. In endothelial cells, H_2_O_2_ (125–1000 μM) stimulates the activity of ^•^NO synthase, contributing to oxidative cellular damage [170,171,172].

Enzymatic antioxidants such as catalase and glutathione peroxidase, along with α-tocopherol, exert inhibitory effects on apoptosis. However, ascorbic and gallic acids have been shown to enhance H_2_O_2_-induced neuroapoptosis. Notably, several antioxidants, including α-tocopherol, exhibit potent antiproliferative properties. In the presence of metal ions with varying valence states, ascorbic acid can display pro-oxidant characteristics and augment H_2_O_2_-induced neuroapoptosis [173,174,175]. Furthermore, it is noteworthy that ^•^NO production can have significant negative effects on the macrophages producing it. Studies have demonstrated that phagocytosis and the production of reactive oxygen species are markedly inhibited in rat or peritoneal macrophages cultured under conditions conducive to ^•^NO production. Macrophages expressing inducible iNOS or treated with ^•^NO have condensed nucleus and cytoplasm. Consequently, the release of ^•^NO by activated macrophages results in their functional suppression and eventual apoptosis. These observations are directly linked to ^•^NO, as they are effectively prevented by the addition of NOS inhibitors. However, an additional challenge in investigating the mechanisms of ^•^NO cytotoxicity is related to the ^•^NO donors used for its generation, as described earlier. The utilization of S-nitrosothiols, particularly GSNO (S-nitrosoglutathione) and SNAP (S-nitroso-N-acetylpenicillamine), in numerous studies raises concerns regarding their potential involvement in transnitrosylation reactions. These reactions entail the transfer of NO^+^ groups to thiols, including glutathione and sulfhydryl groups of proteins, thereby disrupting their cellular functions. The attribution of such reactions to the effects of ^•^NO itself remains ambiguous [176,177,178,179,180,181,182] Importantly, studies have indicated that ^•^NO, whether produced endogenously by macrophages or administered exogenously, primarily inhibits oxidative phosphorylation in the mitochondria of neurocytes. This occurs because ^•^NO reversibly binds to cytochrome oxidase in the electron transport chain of mitochondria. Furthermore, there is evidence suggesting that ^•^NO can directly activate the opening of giant pores, resulting in the release of cytochrome C and the initiation of the caspase cascade, as previously described. Additionally, ^•^NO and its derivatives have the capacity to induce the peroxidation of phospholipids and the oxidation of thiol groups present on mitochondrial membrane proteins. These processes ultimately contribute to the release of apoptogenic factors into the cytosol [183,184,185,186,187].

### 3.2. Anti-Apoptotic Effects of ^•^NO

In addition to the extensively described cytotoxic effects of ^•^NO, numerous studies in the literature have highlighted ^•^NO cytoprotective actions. However, upon comparing the evidence for the antiapoptotic effects of ^•^NO with the cytotoxic actions outlined earlier, it becomes evident that there are contradictions on many fronts [188,189]. This discrepancy underscores the highly ambiguous nature of ^•^NO’s actions, which are contingent upon various conditions.

The molecular mechanisms underlying the antiapoptotic effects of the ^•^NO-mediated expression of HSP may involve two potential pathways [190]. The first possibility entails the direct suppression of apoptotic signal transduction pathways, involving the inhibition of caspase family protease activation. The second involves the chaperone-mediated import of precursor proteins into mitochondria by HSP. This action controls mitochondrial function and membrane permeability, thereby preventing the release of cytochrome C, which is required for further activation of caspases. The relationship between Hsp70 and the induction of apoptosis in obstructive nephropathy was first discussed [191,192]. Other results have shown that Hsp70 can modulate the apoptosis cascade during renal obstruction [193]. Recently, we reported that nitric oxide prevents obstruction-induced cell death through the mitochondrial apoptotic pathway via the induction of heat shock protein 70 [194,195]. Our results demonstrated that the apoptotic effect induced by decreased levels of nitric oxide led to a reduced expression of Hsp70. This was associated with a direct induction of apoptotic signal transduction involving caspase 3 activation by decreasing Bcl-2 stabilization. For some cell types, it has been shown that the protective effect of ^•^NO is mediated through the synthesis of cGMP [196,197]. Moreover, such an effect is produced by rather low doses of ^•^NO, similar to those produced in vivo by ^•^NO synthases. It is assumed that cGMP generation can activate cGMP-dependent protein kinases, which in turn affect proteins of apoptotic cascades (e.g., caspases or Bcl-2) [198,199]. 

The protective effect of low doses of ^•^NO: The pretreatment of macrophage cells with low, nontoxic doses of GSNO (25–200 μM) induced resistance to higher doses of GSNO (1 mM) upon repeated exposure [200,201]. Similarly, the pretreatment of macrophages with LPS and IFN-γ in the presence of L-NMMA induced a comparable effect. Thus, the inducible defense mechanisms that suppress ^•^NO-induced apoptosis are activated by the action of ^•^NO-releasing substances as well as through pre-activation by lipopolysaccharides or cytokines. In other studies, low doses of ^•^NO were found to delay the opening of the giant pore and subsequent apoptotic events [202,203,204].

^•^NO and defense proteins: Several studies have demonstrated the enhanced expression of heat shock proteins (HSP) and Bcl-2 family proteins in response to ^•^NO. Heat shock proteins serve to protect the cell from various stressors, primarily temperature increases, but also oxidative stress and cytokine-induced cytotoxicity. The protective effect of ^•^NO was observed when hepatocytes were treated with tumor necrosis factor and when cells were deprived of serum [205,206,207]. In both cases, the anti-apoptotic action of ^•^NO correlated with an increase in Hsp70 synthesis. Hsp70 is characterized by its function as a molecular chaperone, assisting in protein folding and the removal of damaged proteins.

Hsp70 is known to play a crucial role in protecting the cell from ROS and mitochondrial damage by suppressing the interaction of proteins that transmit death signals to the mitochondria. Bcl-2, a proto-oncogene with anti-apoptotic properties, has already been described above. Increasing its expression upon ^•^NO treatment prevents apoptosis, most likely through the inhibition of giant pore opening [208,209,210]. A novel alternative anti-apoptotic mechanism of ^•^NO involves the induction of Hsp32 (hemoxygenase) and Hsp70 through the ^•^NO-mediated modification of intracellular antioxidant levels [190,211]. The mechanism by which ^•^NO stimulates Hsp70 expression may involve the interaction of ^•^NO with thiol-containing molecules. There is ample evidence indicating that ^•^NO readily oxidizes low molecular weight thiols to form S-nitrosothiols and disulfides [212,213]. 

Among cellular low molecular weight thiols, glutathione is the most abundant and is also one of the intracellular targets of ^•^NO. ^•^NO can oxidize intracellular-reduced glutathione, thereby altering antioxidant levels within the cell and leading to oxidative or nitrosative stress. This action stimulates the induction of the heat shock proteins Hsp32 (hemoxygenase) and Hsp70, which protect cells from apoptotic cell death induced by tumor necrosis factor (TNF) plus actinomycin D and oxidative or nitrosative stress. The pretreatment of hepatocytes with ^•^NO has been shown to alter the redox state accompanied by glutathione (GSH) oxidation and the formation of S-nitrosoglutathione [214,215,216]. GSH-oxidizing agents (diamide) and GSH-alkylating agents (N-ethylmaleimide) induced Hsp70 mRNA expression, whereas the GSH synthesis inhibitor (buthionine sulfoximine) did not; this suggests that ^•^NO induces Hsp70 expression via GSH oxidation [217,218]. The aforementioned induction may occur via the activation of heat shock. The accumulation of misfolded proteins triggers the mobilization of HSP, leading to the formation of a free pool of Hsp70 and the subsequent removal of the negative regulatory effect on HSF activation during heat shock or other stresses. The released HSF is phosphorylated and assembled into trimers, acquires DNA-binding activity, and leads to an increase in Hsp70 mRNA transcripts. During ^•^NO stimulation, multiple and complex pathophysiologic changes occur in the smooth muscle cells of blood vessels, including protein damage or modifications due to the cytotoxic action of ^•^NO [219,220]. 

## 4. ^•^NO in Health and Disease: Interactions, Clinical Relevance, and Therapeutic Implications

### 4.1. ^•^NO and Superoxide Anion

Both ^•^NO and O^2−^ are significant mediators of inflammation. Activated macrophages are known to release both ^•^NO and O^2−^. It is generally believed that the interaction of these radicals produces the even more cytotoxic peroxynitrite [221,222]. However, there is evidence suggesting that the co-incubation of cells with ^•^NO and O^2−^ results in a cross-protective effect, whereas separately both radicals cause apoptosis or necrosis [223,224]. It is thought that in this case, ^•^NO acts as a scavenger of O^2−^, neutralizing its negative effects. Probably, the protective effect requires a balanced presence of ^•^NO and O^2−^ and a certain redox state of the cell, as it is necessary to neutralize the formation ONOO^−^, which is very likely in this situation [225]. Under normal physiological conditions, a balance between superoxide and nitric oxide exists in vivo. ^•^NO and superoxide react together at a diffusion-controlled rate to form peroxynitrite (ONOO^−^), which causes cellular damage by oxidizing many biological molecules. Additionally, ONOO^−^ is involved in the inactivation of Mn and Fe superoxide dismutase [226,227]. ^•^NO can protect cells from cytotoxicity, ROS-mediated by removing superoxide anions, which are involved in toxicity through the formation of hydrogen peroxide or hydroxyl radicals [228]. Nitric oxide has been shown to inhibit the formation of superoxide anions. The mechanism of this inhibition is thought to be due to the inactivation of nicotinamide adenine dinucleotide phosphate oxidase because of the scavenging action of ^•^NO on superoxide [229]. 

The inhibition of caspases: Since cysteine is present in the active center of caspases, and reactive nitrogen species can nitrosylate SH-groups, the initial explanation for the suppression of the caspase cascade by nitric oxide was through such nitrosylation of functionally important Cys, showing not only the suppression of active caspases by nitric oxide but also the interruption of caspase activation itself. The proteolytic activation of caspases 3 and 8 was found to be effectively inhibited by both endogenous and exogenous ^•^NO, and part of this inhibition was unrelated to S-nitrosylation [230,231].

### 4.2. ^•^NO and Arterial Hypertension

In studies on rats with spontaneous hypertension, it has been found that the central component of ^•^NO-ergic regulation of blood pressure involves neurons located in various regions of the brain, including the hypothalamus (such as the paraventricular and supraoptic nuclei, as well as the median eminence) and the medulla oblongata (including the nucleus of the solitary tract, dorsal nucleus, and ambiguous nucleus). Additionally, the peripheral component comprises ^•^NO-producing vascular endothelial cells and neurons in the adrenal medulla.

The development of arterial hypertension is accompanied by specific changes in the activity of ^•^NO-ergic neurons in the brain involved in blood pressure regulation. These changes include a decrease in the number of neurons positive for neuronal nitric oxide synthase (NOS) in the small cell zone of the paraventricular nucleus, fibers of the median eminence of the hypothalamus, and neurons of the nucleus of the solitary tract. Conversely, there is an increase in the number and activity of NOS-positive neurons in the endocrine nuclei of the hypothalamus, as well as the dorsal and ambiguous nuclei of the medulla oblongata. The systemic increase in blood pressure in spontaneous hypertension leads to the inhibition of ^•^NO-producing function in the endothelium of both muscular and elastic vessels. Additionally, the change in ^•^NO-ergic activity in adrenal medullary neurons exhibits a dynamic character. Our studies also revealed a depression of ^•^NO formation alongside a decrease in total nitric oxide synthase (NOS) activity, both in mitochondria and in the cytosol of the myocardium in all groups of SHR [116]. 

We observed a significant increase in the expression of inducible nitric oxide synthase (iNOS) in the myocardial mitochondria of SHR rats compared with normotensive animals. The discoordination between the activity of total NOS in mitochondria and the formation of stable ^•^NO metabolites in the myocardium under conditions of experimental atherosclerosis alongside arterial hypertension, in our opinion, is associated with a surge of “parasitic” reactions. These reactions occur when NOS produces not only ^•^NO but also its cytotoxic derivatives, such as peroxynitrite and the nitrosonium ion, etc. Such reactions may occur in conditions of L-arginine deficiency, antioxidant deficiency, mitochondrial dysfunction, increased iNOS expression, and under the influence of proinflammatory factors [232]. Our assumption is confirmed by the detection of increased content of the nitrosative stress marker nitrotyrosine against the background of increased iNOS expression in the mitochondrial fraction of SHR heart homogenate. Additionally, in the myocardial cytosol of SHR rats, we observed a low level of stable ^•^NO metabolites (1.6–2.4 times lower) compared with normotensive rats, alongside the inhibition of endothelial nitric oxide synthase (eNOS) activity. Analyzing the obtained results of studies on the ^•^NO system parameters and reduced intermediates of the thiol–disulfide system, we can conclude that in SHR rats with the most pronounced shifts of the myocardial thiol–disulfide system (including deficit of reduced equivalents, increased oxidation of intermediates, and deprivation of glutathione reductase activity), there were significant changes in the neurochemical profile of ^•^NO. It transitioned from a molecular messenger to an agent of nitrosative stress [138]. We have demonstrated that arterial hypertension is accompanied by the inhibition of NOS activity and ^•^NO deficiency. This deficiency, combined with the corresponding redox status of mitochondria, leads to protective effects that increase the cell’s resistance to adverse effects. In this context, the expression of inducible NOS increases in mitochondria, particularly when arterial hypertension is combined with diabetes and atherosclerosis. This increased expression has a compensatory value aimed at reducing blood pressure. However, under conditions of deficiency in reduced equivalents in the thiol–disulfide system of cardiac mitochondria, inducible NOS appears as an initiator of nitrosative stress. In this regard, it is important to determine the factor that determines whether ^•^NO exhibits cytoprotective or cytotoxic properties at a certain stage of the molecular–biochemical cascade. The thiol–disulfide system seems to play a special role in the development of mechanisms underlying ^•^NO cytotoxicity and target organ damage. Intermediates of the thiol–disulfide system possess transport properties with respect to ^•^NO, thereby increasing its bioavailability. Moreover, many thiols, such as glutathione, cysteine, and methionine, can significantly limit the cytotoxicity of ^•^NO and its derivatives, thus reducing the degree of damage to the target organ [138,182,233]. 

### 4.3. ^•^NO and the Thiol–Disulfide System of Neurons

The addition of CDNB (80 μmol), a selective inhibitor of glutathione-S-transferase and a glutathione conjugate, to the incubation medium of neurons resulted in the depletion of the glutathione linkage of the thiol–disulfide system (TDS), as evidenced by the deficiency of reduced forms of glutathione due to the inhibition of glutathione reductase (GR) and glutathione-S-transferase (G-S-T) activity. This depletion leads to the uncontrolled production of reactive oxygen species, nitrogen, and nitrosative stress, as indicated by the observed increase in the level of nitrotyrosine in the neuronal suspension [182,234]. 

Thus, the increase in nitrotyrosine in neurons treated with CDNB was found to be more than 2.2-fold. Concurrently, there was a shift of the TDS towards oxidized thiols, as evidenced by a decrease in the level of reduced glutathione by 6.6-fold and an increase in its oxidized form by 3-fold. The accumulation of glutathione disulfide proceeded against a background of decreased activity of key enzymes of TDS: glutathione-S-transferase (G-S-T) decreased by 2.7-fold and glutathione reductase (GR) decreased by 2.3-fold compared with intact neurons at 60 min of incubation. It is important to note that the described pathophysiological changes led to an increase in cellular damage in the neuron suspension, as evidenced by a statistically significant (*p* ≤ 0.05) increase in the number of degenerately changed neurons in the test with Hoechst 33342. A possible mechanism of cell damage in neurons incubated with CDNB, in our opinion, may involve the disruption of the TDS and the formation of mitochondrial dysfunction. It has been established that the deficit of glutathione not only occurs in conditions of the accumulation of active derivatives of ^•^NO but also the decrease in its reduced form can be a triggering factor for the development of nitrosative stress. Restored thiols are intracellular ^•^NO scavengers. Nitric oxide interacts with cysteine to form S-nitrosocysteine and with glutathione to form S-nitroglutathione. S-nitroglutathione serves as the main transport molecule for ^•^NO transfer [138]. The deficiency of sulfhydryl (SH) groups inside the cell leads to a decrease in ^•^NO bioactivity and the accumulation of reactive oxygen species (ROS). Additionally, the uncontrolled growth of ROS leads to the oxidation of the alkyl groups of the mitochondrial respiratory chain and the inactivation of mitochondrial superoxide dismutase (SOD), further depleting the antioxidant system of the neuron. When modeling acute cerebral ischemia in Wistar rats, we observed marked differences in the concentrations of glutathione (GSH) and nitrotyrosine among the groups of animals with mild, moderate, and severe neurological disorders, as reported by McGraw. After conducting statistical analysis using Pearson’s coefficient, a negative correlation of −0.8289 was observed between neurological symptoms and reduced glutathione, while a positive correlation of 0.8272 was found with nitrotyrosine levels. The strong correlations suggest a clear dependence between the studied parameters. Consequently, it appears feasible to compute the ratio of the nitrotyrosine level to the reduced glutathione and utilize it for diagnosing neurological disorders. The calculated coefficients indicate that, under normal conditions, the ratio of nitrotyrosine to glutathione (Kn/GSH) is approximately 1.3. A mild degree of neurological deficit is characterized by a Kn/GSH close to 5.0; in severe neurological disorders, the Kn/GSH ratio increases substantially to about 138.5. Thus, the interaction within the “^•^NO—reduced thiols” system plays a crucial role in the mechanisms of neurodegradation and endogenous neuroprotection, with its ratio determining the fate of neurons under conditions of central ischemia. The key factor of equilibrium in this system is the maintenance of the pool of reduced thiols and, especially, glutathione at a certain level. The reduced glutathione equivalents not only ensure the bioavailability of ^•^NO but also safeguard the proper functioning of the ^•^NO system within neurons, thereby preventing the formation of its neurotoxic derivatives [182]. A statistically significant linear correlation between the severity of the neurological deficit and the functionality of the conjugated “^•^NO—reduced thiols” system was identified. These findings provide experimental support for utilizing the nitrotyrosine/reduced glutathione coefficient as a diagnostic parameter for assessing the severity of cerebral stroke in clinical biochemistry. Testing its effectiveness in treating patients with cerebral blood flow disorders appears promising.

### 4.4. ^•^NO and Cerebral Ischemia

Numerous studies have demonstrated the direct involvement of ^•^NO in the neuronal destruction process during ischemia. This has been observed when selective inhibitors of neuronal and inducible NOS isoforms are administered to animals with acute cerebral circulatory disorders (ACBD), as well as in experiments involving animals with a deficiency in the gene encoding iNOS. Data also indicate an elevation in ^•^NO concentration in the brains of animals experiencing both focal and global ischemia [138]. The concentration of ^•^NO begins to rise within the first minutes of ischemia, peaking on the 1st to 3rd day. Measurements of NOS activity revealed a significant increase in enzyme activity both within the ischemic core and in the penumbra. However, this assessment did not differentiate between the various NOS isoforms. The involvement of ^•^NO in neuronal damage and death exhibits specificity determined by NOS isoforms and the type and stage of stroke development. In the initial phase of ischemia, the expression of constitutive calcium-dependent NOS was triggered by transmitter autocoidosis. ^•^NO production during this phase is not directly responsible for neuronal death but contributes to indirect mechanisms such as the activation of phospholipases, the augmentation of hydroxyl radical formation, and the modulation of NMDA receptor activity. Subsequently, from 7 to 14 days in global ischemia and from 1 to 3 days in focal ischemia, during the delayed post-ischemic period, there is a surge in ^•^NO production involving inducible NOS activated within glia, macrophages, and neutrophils [138]. The delayed induction of inducible NOS expression correlates with the subsequent activation of astro- and microglia as well as inflammatory cells. In focal ischemia, these cells, known as ^•^NO producers, are localized within the penumbra, while in global ischemia, they are primarily found in structures most vulnerable to oxygen deficiency. Apart from ^•^NO synthases, nitrate/nitrite reductases in warm-blooded organisms serve as sources of ^•^NO, capable of reducing nitrate and nitrite. Gliocytes and thymocytes exhibit nitroreductase activity. Although xanthine oxidase has demonstrated the ability to convert nitrate and nitrite into ^•^NO, its role in neurodegeneration remains understudied. Currently, there is active research into the targets of nitric oxide and efforts to elucidate whether ^•^NO itself is sufficiently cytotoxic or if its derivatives are more active [235]. 

It is well-established that ^•^NO within target cells forms active derivatives such as nitrosonium (NO^+^), nitroxyl (NO^−^), and peroxynitrite (ONOO^−^). Recent studies have further emphasized the role of ^•^NO and its transformation products, including peroxynitrite (ONOO^−^), nitrosonium ion (NO^+^), nitroxyl (NO^−^), and diazotrioxide (N_2_O_3_), as primary factors in inducing nitrosative stress [182,236]. This stress arises from both the direct interaction of ^•^NO with metals, such as heme iron in hemoglobin and myoglobin, iron-containing enzymes, and non-heme iron in iron-sulfur proteins and DNA, as well as copper and zinc in enzyme active centers. Additionally, the indirect interaction of NO^+^ through S-, N-, and O-nitrosation with thiol, phenolic, hydroxyl, and amino groups of proteins and DNA further contributes to nitrosative stress. Such interactions lead to receptor desensitization, the inhibition of mitochondrial enzyme activity, and nucleic acid fragmentation. Consequently, ^•^NO, which reversibly binds to the Fe3+ active center of catalase, significantly inhibits its function both during the initial period of ischemia and in the post-ischemic phase of focal cerebral ischemia. Excessive ^•^NO levels depress heme enzymes within the mitochondrial electron transport chain. In the post-ischemic period, elevated ^•^NO concentrations can interact with heme iron and paired thiol groups to form a dinitrosyl iron complex (DNIC) [182]. Unlike ^•^NO, DNIC serves as a potent nitrosylating agent, interacting with protein thiols, histidine, aspartate, glutamine, methionine, cysteine, and glutathione, forming N- and S-nitrosothiols. Under ischemic conditions, DNIC undergoes irreversible nitrosylation of iron-sulfur clusters in mitochondrial proteins (such as NADH-ubiquinone oxidoreductase, succinate-ubiquinone oxidoreductase, and aconitase), thereby contributing to mitochondrial dysfunction [182]. Our research has demonstrated that DNIC significantly inhibits the activity of superoxide dismutase (SOD), as well as the enzymes involved in regulating thiol–disulfide equilibrium within cells, including glutathione reductase, glutathione-S-transferase, and glutathione peroxidase in neuronal suspensions.

Under ischemic conditions, the inhibition of these enzymes leads to the oxidative modification of low-molecular-weight thiols, resulting in the formation of homocysteine and the subsequent impairment of ^•^NO transport. This impairment leads to the generation of cytotoxic derivatives of ^•^NO, which further exacerbate thiol oxidation. Neurons equipped with a sufficiently active thiol antioxidant system capable of regulating ^•^NO transport exhibit resistance to nitrosative stress, which represents the earliest neurodegradative mechanism under ischemic conditions. It is well-documented that within the initial minutes of brain ischemia, ^•^NO (whether macrophage-derived or exogenous) inhibits oxidative phosphorylation in the mitochondria of target cells through reversible binding to mitochondrial cytochrome-C oxidase. The suppression of electron transport in mitochondria leads to the generation of superoxide, resulting in the formation of ONOO^−^. Subsequently, peroxynitrite synthesis occurs in cells with high activity of ^•^NO synthase and enzymes producing ROS (such as xanthine oxidase, NADH oxidoreductase, cyclooxygenase, lipoxygenase, and electron transport chain enzymes). Recent studies have revealed that during the initial stages of ischemia, peroxynitrite levels can be mitigated by mitochondrial nitroreductase, which reconverts it back to ^•^NO using NADPH and NADH as cofactors. The targets of oxidative and nitrosative attacks by peroxynitrite encompass thiols, CO_2_, metalloproteins, nucleic acids, transmitters, and lipids [182]. 

Peroxynitrite, being a relatively stable compound, undergoes rapid protonation to form its primary product, the nitrate anion, along with hydroxyl radicals and nitrogen dioxide, thereby determining its oxidative properties. Hence, during the initial stages of ischemia, peroxynitrite interacts with thiols via nitrosylation, leading to the formation of nitrosothiols. As the process progresses and lactate acidosis ensues, this interaction shifts towards oxidation, resulting in the formation of more persistent disulfides. These reactions significantly contribute to the mechanisms of neurodegradation by shifting the thiol disulfide system towards oxidized thiol compounds, thereby reducing the cell’s reductive potential. This oxidation process also disrupts gene expression by irreversibly oxidizing cysteine residues within redox-dependent domains and causing the dissociation of the MAP kinase cascade. Moreover, peroxynitrite inhibits the activity of metabolic cycles involving methionine and cysteine, thereby impeding key enzymes regulating cysteine levels and promoting homocysteine formation. Additionally, peroxynitrite reacts with the metabolitotropic transmitter CO_2_ to form a potent nitrosylating agent, nitrosoperoxycarbonate. An essential mechanism of peroxynitrite’s neurotoxic action is its reaction with thiosin to form nitrotyrosine. Peroxynitrite significantly inhibits the activity of Cu-Zn-SOD and Mn-SOD by nitration of its 34th tyrosine residue and by binding to copper, altering its valence. Moreover, peroxynitrite serves as a specific agent that irreversibly depresses mitochondrial respiration during ischemia (Figure 2). Direct interaction with the iron of active centers of key enzymes and the nitrosylation of thiol, phenol, hydroxyl, and amino groups of the protein component of these enzymes by S-, N-, and O-elements, results in their irreversible oxidation under heightened nitrosative stress. The suppression of mitochondrial respiration leads to a decline in mitochondrial charge, which can trigger the apoptotic process and, in the absence of glucose, necrosis [6,237,238]. Evidence also suggests the direct activation of the giant pore opening by nitric oxide, leading to the release of cytochrome C and the triggering of the caspase cascade. These findings were obtained when mitochondria were exposed to cytotoxic derivatives of ^•^NO such as peroxynitrite and nitrosonium ion, whose mechanism is based on the modification of thiol proteins in the mitochondrial pore.

^•^NO and its derivatives can induce the peroxidation of phospholipids. Consequently, cytotoxic derivatives of ^•^NO, along with hydroxyl radicals, trigger the opening of mitochondrial pores and the expression and release of proapoptotic proteins into the cytosol. This pore opening occurs due to the oxidation or nitrosylation of thiol groups within the cysteine-dependent portion of the mitochondrial inner membrane protein, specifically the ATP/ADP-antiporter, transforming it into a permeable nonspecific channel-pore. This transformation converts mitochondria from “power plants” into “furnaces” of oxidation substrates without ATP formation [182,239]. Impaired tissue oxygenation, transmitter autocoidosis, disrupted calcium accumulation by mitochondria, and damage to the mitochondrial membrane by cytotoxic ROS and ^•^NO compounds further enhance pore opening, leading to the release of apoptogenic proteins from damaged mitochondria [240]. The mitochondrial pore is a channel spanning both mitochondrial membranes and comprises three proteins: an adenine nucleotide translocator, a potential-dependent anion channel (porin), and a benzodiazepine receptor. When this complex binds to Ca^2+^, substances with small molecular weight can traverse the membrane pore. This results in a reduction in membrane potential and the swelling of the matrix, ultimately compromising the integrity of the outer membrane and leading to the release of apoptotic proteins from the intermembrane space into the cytoplasm.

The nitrosylation of proteins by tyrosine residues, facilitated by ONOO−, can have significant functional consequences, as it suppresses tyrosine phosphorylation and disrupts certain signal transduction pathways within the cell [182,241]. The balance between ^•^NO and the thiol–disulfide system is a critical factor determining the subsequent fate of neurons under ischemic conditions, particularly the mode of cell death. During ischemic brain injury, nitrosative stress emerges early, leading to thiol nitrosation and altering the thiol–disulfide equilibrium of mitochondrial pore proteins. At this juncture, mitochondrial NOS assumes a protective role by modulating cell death, favoring a transition towards apoptosis. Subsequently, oxidative and carbonyl stress ensue, resulting in a significant shift in the thiol–disulfide equilibrium towards oxidized thiols. This leads to persistent mitochondrial dysfunction, the depletion of cellular energy reserves, the onset of autocoidosis, the perturbation of genomic responses, and ultimately, cell death via necrosis [242]. 

### 4.5. ^•^NO and Endothelial Dysfunction

The primary mechanism underlying endothelial dysfunction (ED) involves a reduction in the formation and bioavailability of ^•^NO, accompanied by a concurrent increase in the level of superoxide ions and the production of active vasoconstrictors [243,244]. Consequently, ED manifests as an imbalance between mediators crucial for the optimal functioning of all endothelium-dependent processes under normal conditions [245]. Concurrently, disruptions in the production, interaction, and breakdown of endothelial vasoactive factors are observed, alongside abnormal vascular reactivity and alterations in the structure and growth of blood vessels, which are indicative of vascular diseases [246]. 

^•^NO is synthesized from L-arginine under the influence of endothelial ^•^NO synthase (eNOS), a process involving the attachment of molecular oxygen to the terminal nitrogen atom of the guanidine group of L-arginine. The assessment of vascular wall integrity and the correction of ED in cardiovascular pathology represent one of the most promising fields of study, as they determine the likelihood of developing vascular diseases and their complications, thus contributing to the overall prognosis of the disease [247]. 

Therefore, the pursuit of targeted interventions for Endothelial Dysfunction (ED) and the development of a new class of effective drugs—endothelium protectors—represent critical clinical and experimental endeavors. ED is a systemic pathology linked to the compromised microstructure and secretory function of endothelium-dependent cells, resulting in reduced endothelium-dependent vasodilation, hypercoagulability, increased thrombosis, heightened vascular permeability, and lipoprotein migration into the vascular intima, as well as smooth muscle cell proliferation, and myocardial and vascular remodeling [248,249,250]. The primary mechanism underlying the development of ED involves a reduction in the formation and bioavailability of ^•^NO, accompanied by the emergence of its cytotoxic forms amidst oxidative stress and a deficiency of reduced low molecular weight thiols [98,244]. Meanwhile, the primary causes of ^•^NO deficiency in endothelial cells may include a reduced content of its precursor, L-arginine, the diminished expression or activity of endothelial nitric oxide synthase (eNOS), and a deficiency in ^•^NO synthesis cofactors, particularly tetrahydrobiopterin. Additionally, increased levels of endogenous eNOS inhibitors, such as asymmetric dimethylarginine and monomethyl-L-arginine, the elevated formation of reactive oxygen species, notably superoxide anion, and the presence of low-density lipoproteins, especially their oxidized forms, contribute to ^•^NO depletion [251,252]. The molecular basis of vascular endothelial dysfunction remains complex and not entirely understood. However, the “eNOS—L-arginine—^•^NO” system holds promise as a pivotal target for the pharmacological correction of ED in the foreseeable future [98,253]. Numerous authors have highlighted the direct involvement of ^•^NO in cell death processes, including endothelial cells, under conditions such as ischemia, atherosclerosis, and alcohol intoxication. These findings were elucidated through the utilization of selective inhibitors targeting constitutive and inducible isoforms of nitric oxide synthases (NOS), alongside experiments conducted on animals with a deficiency in the gene encoding inducible NOS (iNOS). Investigations have demonstrated that ^•^NO transport occurs concomitantly with the formation of N_2_O_3_, subsequently leading to thiol nitrosylation. With the involvement of disulfide isomerase, ^•^NO is released [234]. Additionally, there exists a mechanism for ^•^NO release from S-nitrosoglutathione, facilitated by glutamyl transpeptidase, resulting in the formation of S-nitrosocysteinylglycine, which then liberates ^•^NO. Cystine plays a crucial role in the transportation of S-nitrosoglutathione, wherein it is reduced to cysteine. The latter, upon reacting with S-nitrosoglutathione, forms S-cysteine, thereby participating in the rapid conduction of neurons and facilitating the neuron’s adaptive responses to ischemia. These reactions are regulated by glutathione reductase and glutathione transferase.

Under ischemic conditions, the inhibition of these enzymes leads to the oxidative modification of low molecular weight thiols, homocysteine formation, and the subsequent impairment of ^•^NO transport, resulting in the generation of cytotoxic ^•^NO derivatives that exacerbate thiol oxidation [254,255,256]. Given the current absence of specific drugs for correcting endothelial dysfunction (ED), insights into the effects of cardiovascular drugs from various pharmacological groups on endothelial functional characteristics hold significant value. A comprehensive approach to treating ED in conditions such as chronic cerebral ischemia, arterial hypertension, alcoholic myocardial and cerebral damage, and chronic heart failure may offer substantial practical benefits [257]. This approach involves combining fundamental cardioprotective and neuroprotective therapies with medications that optimize energy metabolism, thereby mitigating the adverse effects of oxidative and nitrosative stress on vascular endothelium and promoting nitric oxide formation. It is conceivable that the future lies with drugs possessing not only cardioprotective or neuroprotective effects but also indirect positive impacts on endothelial function. A particularly promising avenue is the comprehensive treatment of ED in cardiovascular pathology, where reperfusion, antithrombotic, and cardio- or neuroprotective therapies are integrated with medications targeting endothelial dysfunction correction [258].

In light of the above, it has become pertinent to investigate the endothelioprotective properties of drugs exhibiting diverse pathogenetic mechanisms of action. These drugs are known to enhance metabolism, possess antioxidant properties, serve as natural ^•^NO donors, and activate the ^•^NO synthase enzyme. They also contain “essential” phospholipids and affinity-purified antibodies to endothelial ^•^NO synthase, offering promising avenues for research in experimental models of cerebrovascular pathology [259]. 

Numerous drugs with distinct mechanisms of action exert varying degrees of influence on vascular endothelial function. For instance, nitrates replenish endogenous ^•^NO deficiency, and ACE inhibitors not only reduce angiotensin-II (AT-II) synthesis but also prevent kinin degradation. Statins bolster the endothelial cell barrier function against oxidized LDL, while calcium antagonists curb AT-II and endothelin activity in vascular smooth muscle, thereby amplifying ^•^NO’s vasodilatory effects. Angiotensin receptor blockers obstruct AT-II receptors, fostering ^•^NO accumulation, while endothelin-converting enzyme inhibitors and endothelin-1 receptor antagonists impede peptide activity [260]. 

Of particular interest are the “specific” effects directed at enhancing ^•^NO synthesis, such as replacement therapy involving L-arginine (the substrate eNOS) and tetrahydrobiopterin (eNOS cofactor), crucial for determining the enzyme’s activity [261].

### 4.6. Pharmacological Modulation of the Nitroxidergic System

Pharmacological modulation of the nitroxidergic system is presented in Table 1.

### 4.7. Inhibitors of NOS Isoforms and Their Cytoprotective Effect of Neurons

In experimental cerebral ischemia, the application of NOS inhibitors with varying selectivity leads to complex changes. For instance, L-NAME, a non-selective inhibitor, induces the irreversible inhibition of constitutive and reversible inducible isoenzyme activity, demonstrating prooxidant properties during the early stages. L-NAME’s inhibition of eNOS compromises local vasodilation, exacerbating the overall pathological process [262]. 

N-propyl-L-arginine during the first 12 h of ischemia caused a significant decrease in ^•^NO level, but the effect on other parameters did not reach statistically significant differences. The mentioned inhibitor is selective in relation to nNOS, the activity of which is significantly increased during the first hours of ischemia. This is explained, firstly, by the observation period—by 12 h, the hyperactivation of nNOS caused by calcium ions begins to decrease with a parallel increase in the inducible form, and secondly, the inhibition of nNOS leads to the activation of the nuclear factor NF-kB, which causes the induction of iNOS [263]. 

Treatment with the nNOS inhibitor—N-propyl-L-arginine—on the 1st and 4th day did not significantly affect the studied parameters, since in more delayed periods the contribution of this isoenzyme to the formation of nitrosative stress was not significant. ^•^NO hyperproduction at these stages is caused by the participation of iNOS in glial cells, macrophages, and neutrophils. The delayed nature of the iNOS increase is associated with the later activation of astroglia. In contrast to nNOS and eNOS, iNOS remains active for a longer period of time and produces significant concentrations of ^•^NO [234]. This explains the beneficial effect of the inhibitors we found, which selectively suppress the activity of the inducible isoenzyme late in the observation period. At the end of 1 day after the modeling of cerebral ischemia, the administration of (S)-methylthiourea caused a significant decrease in the manifestations of nitrosative stress, its effect was more prolonged, and they persisted until the end of observation. On the 4th day of the experiment, the indicated drug reduced the level of protein oxidative modification products, nitrotyrosine and MDA [234]. 

The application of L-NAME caused similar but less pronounced changes, which is apparently associated with the inhibitory effect of this compound on eNOS. Our studies have noted the role of intermediates and enzymes of the thiol–disulfide system in the mechanisms of ^•^NO bioavailability both in in vitro experiments and in modeling cerebral ischemia in rats [234]. Decreased activity of enzymes in the glutathione system, primarily GPO, which ensures the cleavage of nitrosothiols with the release of ^•^NO, in conditions of oxidative stress is one of the reasons for the decrease in its bioavailability. Thus, the neurotoxic effect of ^•^NO depends on a certain NOS isoenzyme [234]. An analysis of the obtained data indicates the limited role of neuronal isoform. The most suitable target for the pharmacological regulation of ^•^NO-dependent mechanisms of neurodegradation is iNOS, as its activity increases 12 h after the development of ischemia, and its action is realized during the next few days.

### 4.8. Exogenous ^•^NO

Exogenous ^•^NO donors are of considerable help in studying the effect of ^•^NO on cells. These substances are widely used nowadays to create model systems in vitro, on which it is possible to study the effects of ^•^NO influence on cultures of different cells, or on separate compartments of cells (isolated mitochondria, nuclei) [264]. These models are very popular because they greatly simplify the system of interaction between ^•^NO and cells in the body. Since nitric oxide here comes from outside, the system appears to be independent of ^•^NO synthases and their regulation, which means that the results of ^•^NO action are easier to interpret. It is evident that in this context, the effects of other signaling substances potentially accompanying biogenic ^•^NO are largely eliminated. For instance, NOS can synthesize O^2−^ under certain conditions, and the pathways activating NOS may also trigger the production of various additional mediators [117,265,266,267]. Exogenous donors, in contrast to L-Arg, incorporate ^•^NO within the structure of the molecule, facilitating the release of this molecule in its pure form [179]. 

Chemical classifications of ^•^NO donors typically include the following groups:-Nitrates (such as nitroglycerin, sodium nitroprusside, and nitrosorbide, commonly used in the clinic) [268,269,270,271,272],-Nitrites (amyl nitrite, NaNO_2_),-Nitrosothiols and substances that form various complexes with ^•^NO: S-nitrosoglutathione (GSNO), S-nitroso-N-acetylpenicillamine (SNAP), diethylamine-NO (DEA-NO).

^•^NO donors can be classified based on their mechanism of action into those that spontaneously release ^•^NO (non-enzymatic) and those requiring enzymatic interaction to release nitric oxide. Currently, there are no “ideal” ^•^NO donors for research purposes. Firstly, ^•^NO donors vary in their efficiency of ^•^NO release and their ability to affect cells to different extents. Secondly, these substances may serve as sources of side compounds, some of which can be toxic (such as cyanide released by nitroprusside) [269,273,274]. 

Among the potential ^•^NO donors and drugs, there is a promising original molecule called bromide 1-(β-phenylethyl)-4-amino-1,2,4-triazolium (Hypertril), a derivative of 1,2,2,4-triazole. Hypertril exhibits ^•^NO-mimetic properties, particularly when a β1-receptor blockade is present. It enhances the expression and activity of endothelial ^•^NO-synthase, thus addressing ^•^NO deficiency. In the dose range of 7.5–20 mg/kg, “Hypertril” shows promising effects in mitigating disorders in the L-arginine-^•^NO -synthase—^•^NO system in spontaneous arterial hypertension. It achieves this by increasing ^•^NO production through the enhanced expression of endothelial NOS, thereby reducing manifestations of nitrosative stress in the myocardium. Additionally, “Hypertril” reduces the expression of inducible NOS, leading to dose-dependent increases in cardiomyocyte nuclei density and cardiomyocyte area. Furthermore, it significantly increases RNA content in both the nuclei and cytoplasm of cardiocytes, along with an increase in the nuclear-cytoplasmic index, indicative of decreased myocardial hypertrophy. Importantly, “Hypertril” also normalizes blood pressure [232,275]. Moreover, the administration of “Hypertril” to animals with chronic heart failure (CHF) results in the prolongation of the depolarization phase (QRS complex) and the repolarization phase of ventricles (T wave), as well as electrical diastole (TR interval). These findings suggest a crucial property of the drug in CHF therapy, specifically its ability to prevent the development of diastolic dysfunction [276]. The obtained results of the experimental studies are the basis for authorization of the first phase of clinical trials of the new drug “Hypertril” as an antianginal and antihypertensive agent.

## 5. ^•^NO Scavengers

### 5.1. Xanthine Derivatives

The high efficacy of 8-benzylaminoxanthines in ROS inhibition assays stems from the ^•^NO radical’s high reactivity and the presence of a secondary amino group in these compounds. These substituents readily undergo nitrosation reactions to produce corresponding N-nitrosoamino derivatives. Pharmaceutical analyses often utilize this interaction to qualitatively confirm the presence of such groups in drug structures [277]. 

It is known that these substituents easily undergo a nitrosation reaction to form the corresponding N-nitrosoamino derivatives. This interaction is used in pharmaceutical analysis to qualitatively confirm the presence of this group in the structure of drugs.

The ^•^NO radical is a potent nitrosating agent, demonstrating its detrimental impact on thiol and amino groups within protein molecules. 8-benzylaminooxanthines can function as ^•^NO scavengers, transforming into corresponding 8-N-nitrosobenzylaminooxanthines under ^•^NO radical exposure, thereby mitigating its adverse effects (Figure 3).

However, this interaction alone does not fully account for the high antioxidant activity observed with 8-benzylaminoxanthines in other in vitro methods. It is important to note that the methylene group of the benzylamine fragment possesses considerable mobility due to the electron-accepting properties of the nitrogen atom and the benzene ring. Consequently, this site can undergo oxidation via dehydration reactions, functioning as a hydrogen atom donor. Thus, the heightened antioxidant activity of 8-benzylaminoxanthines may arise from their capability to oxidize and form corresponding imidazole derivatives or to undergo hydroxylation to produce corresponding hydroxy derivatives (Figure 4) [278,279]. 

The hydrazide 8-benzylaminotheophyllinyl-7-acetic acid (C-3) exhibited the most pronounced antioxidant effect due to its ability to bind ^•^NO via the presence of a hydrazine group in its structure, thereby acting as a spin trap. The administration of C-3 to animals with intracerebral hemorrhage (ICH) resulted in a significant decrease in the expression of neuronal nitric oxide synthase (nNOS) mRNA in the CA1 zone of the hippocampus by 95.3% compared with the control values, along with an increase in nNOS mRNA expression relative to sham-operated animals. Moreover, the expression of inducible nitric oxide synthase (iNOS) mRNA decreased % following C-3 administration relative to the control and was at levels comparable to those of sham-operated animals [280,281,282]. Furthermore, the course administration of compound C-3 to animals with ICH led to a significant decrease in the activity of NOS, nitrites, and nitrotyrosine in brain mitochondria on the 4th day of the experiment, respectively. The administration of C-3 also decreased the expression of iNOS in brain mitochondria. Additionally, C-3 increased the level of HSP70 in the brain cytoplasm and in the mitochondria of animals with ICH [283]. 

### 5.2. 1,2,4-Triazole Derivatives

The compound (S)-2,6-diaminohexanoic acid 3-methyl-1,2,4-triazolyl-5-thioacetate (Angiolin) demonstrates ^•^NO scavenger properties. The ^•^NO activity of this compound is attributed to the reactivity of both its cationic and anionic parts. Specifically, lysine interacts with ^•^NO through its ε-amino group, forming the corresponding N-nitroso derivative. Concurrently, the anionic portion of Angiolin appears to generate S-nitro derivatives, similar to those described elsewhere. It is evident that the ^•^NO activity of both the anionic and cationic parts of Angiolin is synergistic, explaining its pronounced effectiveness. The mechanism of interaction between the Angiolin molecule and ^•^NO may involve electron transfer from the highest occupied molecular orbital of the “spin trap” to the lower unoccupied molecular orbital of the nitrogen monoxide radical, leading to the formation of a more stable complex compound. Angiolin may function as a ^•^NO transfer molecule, enhancing its bioavailability [284].

The administration of Angiolin (100 mg/kg) in chronic cerebral ischemia results in increased survival of endotheliocytes in the vessels of the cerebral cortex and the vascular wall of cerebral vessels. Additionally, it augments the number of proliferating endotheliocytes and enhances the expression of vascular endothelial growth factor (VEGF). Angiolin exhibits the ability to normalize the eNOS/iNOS ratio, as evidenced by a histoimmunohistochemical study of the CA1-hippocampus. Additionally, Angiolin reduces the intensity of nitrosative stress in the ischemic brain, as indicated by a decreased level of nitrotyrosine. Furthermore, it enhances the expression of endogenous neuroprotective agents, such as heat shock proteins (HSP70), observed in both the cytosol and mitochondria of neurocytes. Interestingly, Angiolin also appears to mitigate mitochondrial dysfunction, as evidenced by a decrease in the number of mitochondria exhibiting signs of ultrastructure disorders (Figure 5) [182].

## 6. Conclusions

Thus, despite intensive studies on apoptosis, a detailed understanding of the pathways regulating this process still requires further clarification. It is becoming increasingly apparent that various mechanisms regulating cell death are intricately intertwined, making it challenging to distinguish between pro- and anti-apoptotic components in the actions of signaling molecules. Nitric oxide serves as a prime example, with the scientific literature presenting nearly equal evidence of both the cytotoxic and protective effects of this compound. This complexity complicates the translation of theoretical advancements into practical applications of such substances, as it is challenging to predict their effects at the multicellular organism level based solely on in vitro studies. Therefore, a comprehensive understanding of the regulation of vital cell functions is constructed through focused investigations into the effects of specific compounds under particular conditions on the development of specific signaling pathways. 

## Figures and Tables

**Figure 1 antioxidants-13-00504-f001:**
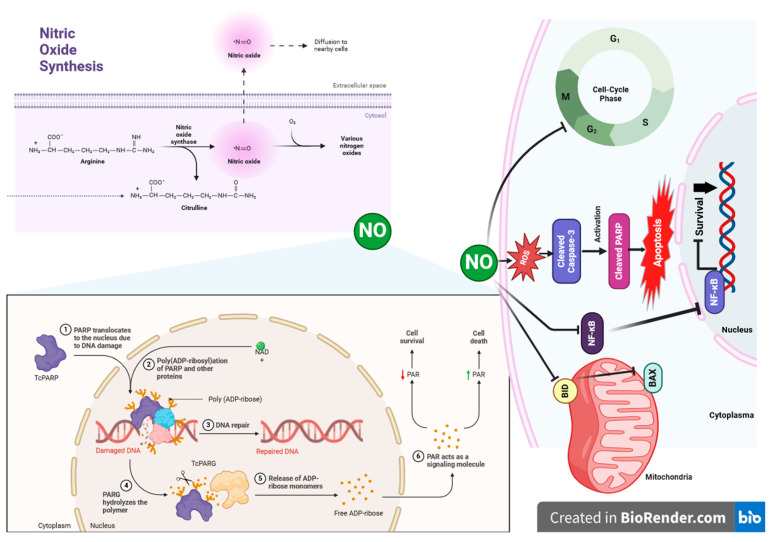
Nitric oxide synthesis and the effect on the PARP system and apoptosis. This figure was generated using BioRender.

**Figure 2 antioxidants-13-00504-f002:**
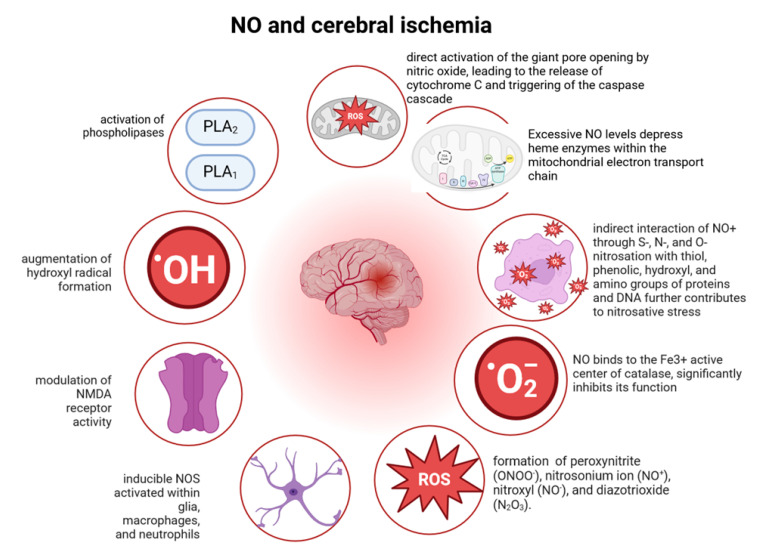
Nitric oxide and cerebral ischemia. This figure was generated using BioRender.

**Figure 3 antioxidants-13-00504-f003:**
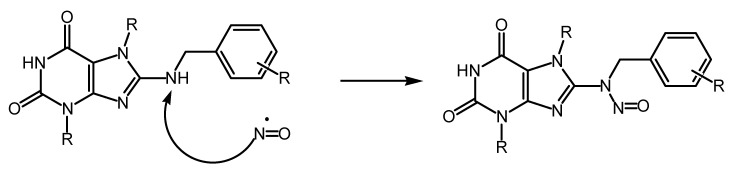
Scheme of the potential interaction of 8-benzylaminoquinones with ^•^NO.

**Figure 4 antioxidants-13-00504-f004:**
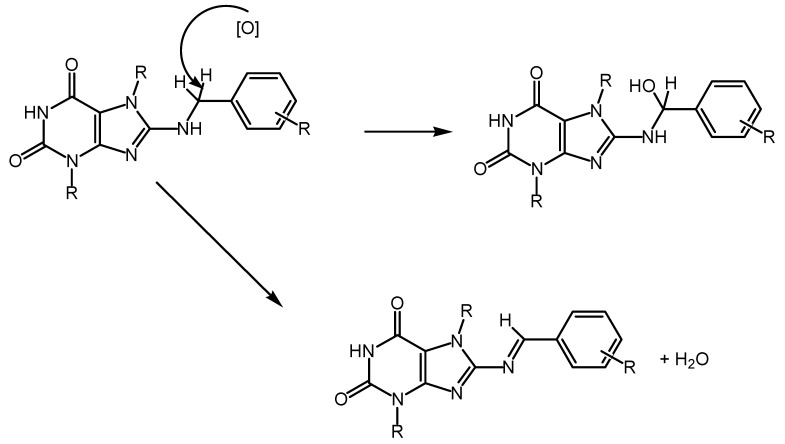
Scheme of the potential oxidation of 8-benzylaminoxanthines.

**Figure 5 antioxidants-13-00504-f005:**
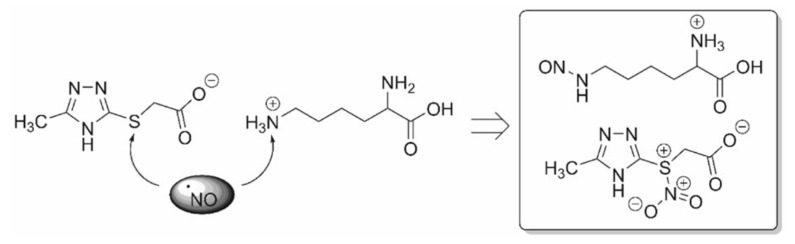
Hypothetical mechanism of (S)-2,6-diaminohexanoic acid 3-methyl-1,2,4-triazolyl-5-thioacetate (Angiolin) interaction with ^•^NO.

**Table 1 antioxidants-13-00504-t001:** Characterization of pharmacological agents—modulators of various links—targets of the ^•^NO system.

Pharmacological Agent	Primary Target	Pharmacological Effect
S-methylisothiourea (SMT)	Selective highly reactive iNOS inhibitor	Injection *w*/*w* to rats after occlusion of carotid arteries (1 mg/kg) over 4 days led to a reliable protective effect only from the 1st day of the experiment, reaching the maximum on the 4th day. SMT had a significant neuroprotective effect [182]
N-nitro-L-arginine methyl ester hydrochloride	Selective iNOS inhibitor	Incorporation of 40 μmol into the neuronal suspension prior to glutamate (100 μM) had a protective effect when incubated for 30 and 60 min (decreased nitrotyrosine, increased GSH, Cu-Zn-SOD) [182,262]
N-propyl-L-arginine hydrochloride	Selective nNOS inhibitor	Incorporation of 50 μmol into the neuronal suspension prior to glutamate (100 μM) had a protective effect when incubated for 30 (decreased nitrotyrosine, increased GSH, Cu-Zn-SOD), then the effect diminished. Injection *w*/*w* to rats after occlusion of carotid arteries (2.5 mg/kg) during 4 days for the first 12 h, a reliable effect [262]

## Data Availability

All the data generated during this research are included in the manuscript.

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
