# Peer review of "Modulating Nitric Oxide: Implications for Cytotoxicity and Cytoprotection"

_antioxidants, 2024, doi:10.3390/antiox13050504_

Round 1

Reviewer 1 Report

In the review paper, the authors have summarized our current knowledge on the role of nitric oxide in modulating cell death and adaptive responses. The review paper is relatively long and comprehensive – the authors have considered 286 references. However, some parts should be re-written. The abstract is too vague and unfocused. Please be more precise. The authors focused mainly on apoptotic cell death. Please note that there are over a dozed of regulated cell death modalities reported. What about other than apoptosis modes of cell death and potential modulatory role of nitric oxide? This issue should be addressed. The authors should focus more on nitric oxide donors/inhibitors (scavengers) as potential drugs. A table summarizing the effects of nitric oxide donors/inhibitors (scavengers) would be useful. The effects of nitric oxide on the activity of selected signaling pathways, namely pro-survival and cell death signaling pathways should be also more emphasized. Furthermore, the authors should take more care about the correct names and formulas of free radicals. For example, nitric oxide is a free radical and the authors did not consequently used the correct formula for nitric oxide in the text. Every free radical should be denoted with a free radical symbol (a superscript dot). Please amend accordingly. Please also change “Ca2+” to “Ca2+”, etc.

In the review paper, the authors have summarized our current knowledge on the role of nitric oxide in modulating cell death and adaptive responses. The review paper is relatively long and comprehensive – the authors have considered 286 references. However, some parts should be re-written. The abstract is too vague and unfocused. Please be more precise. The authors focused mainly on apoptotic cell death. Please note that there are over a dozed of regulated cell death modalities reported. What about other than apoptosis modes of cell death and potential modulatory role of nitric oxide? This issue should be addressed. The authors should focus more on nitric oxide donors/inhibitors (scavengers) as potential drugs. A table summarizing the effects of nitric oxide donors/inhibitors (scavengers) would be useful. The effects of nitric oxide on the activity of selected signaling pathways, namely pro-survival and cell death signaling pathways should be also more emphasized. Furthermore, the authors should take more care about the correct names and formulas of free radicals. For example, nitric oxide is a free radical and the authors did not consequently used the correct formula for nitric oxide in the text. Every free radical should be denoted with a free radical symbol (a superscript dot). Please amend accordingly. Please also change “Ca2+” to “Ca2+”, etc.

Author Response

In the review paper, the authors have summarized our current knowledge on the role of nitric oxide in modulating cell death and adaptive responses. The review paper is relatively long and comprehensive – the authors have considered 286 references. However, some parts should be re-written.

We thank the Reviewer for the suggestions. We have revised the manuscript accordingly.

The abstract is too vague and unfocused. Please be more precise.

We have now revised the Abstract and the Title.

The authors focused mainly on apoptotic cell death. Please note that there are over a dozed of regulated cell death modalities reported. What about other than apoptosis modes of cell death and potential modulatory role of nitric oxide? This issue should be addressed.

The text is now updated and revised.

The authors should focus more on nitric oxide donors/inhibitors (scavengers) as potential drugs. A table summarizing the effects of nitric oxide donors/inhibitors (scavengers) would be useful. The effects of nitric oxide on the activity of selected signaling pathways, namely pro-survival and cell death signaling pathways should be also more emphasized.

The table and a section 4-6 are now included in the text, as suggested. The text is revised.

Furthermore, the authors should take more care about the correct names and formulas of free radicals. For example, nitric oxide is a free radical and the authors did not consequently used the correct formula for nitric oxide in the text. Every free radical should be denoted with a free radical symbol (a superscript dot). Please amend accordingly. Please also change “Ca2+” to “Ca2+”, etc.

The formulas are now corrected.

We thank again the Reviewer for the constructive feedback and comments.

Reviewer 2 Report

This paper provides a detailed review of the importance of NO as a signal molecule. Although authors have covered a wide range of topics, there are some points that would be desirable to correct.

# First of all, part 4 was duplicated.

4. Nitric Oxide (NO) in Health and Disease: Interactions, Clinical Relevance, and 479

Therapeutic Implications

4.NO scavengers

# Also, the subject of 4 “Nitric oxide in health…” was not coherent.

While there were basic subtitles, there were also a mix of clinical topics (arteria hypertension, cerebral ischemia, etc.). In addition, NO also seems to be related to other ischemia, but it seems to be mostly described in terms of cerebral ischemia in the current version.

No comment

Author Response

We thank the reviewer for the constructive comments and feedback.

This paper provides a detailed review of the importance of NO as a signal molecule. Although authors have covered a wide range of topics, there are some points that would be desirable to correct.

# First of all, part 4 was duplicated.

  1. Nitric Oxide (NO) in Health and Disease: Interactions, Clinical Relevance, and 479

Therapeutic Implications

4.NO scavengers

Thank you. We have corrected this point and revised the text accordingly.

 # Also, the subject of 4 “Nitric oxide in health…” was not coherent.

While there were basic subtitles, there were also a mix of clinical topics (arteria hypertension, cerebral ischemia, etc.).

We have now revised and updated the manuscript.

In addition, NO also seems to be related to other ischemia, but it seems to be mostly described in terms of cerebral ischemia in the current version.

The text is now revised and updated.

We thank again Reviewer for the feedback

Round 2

Reviewer 1 Report

The authors have corrected the paper according to my comments. However, some minor revisions are still needed. Please change NO to NO thorough the manuscript. NADH- oxidoreductase should be NADH oxidoreductase. Please check your paper very carefully.

The authors have corrected the paper according to my comments. However, some minor revisions are still needed. Please change NO to NO thorough the manuscript. NADH- oxidoreductase should be NADH oxidoreductase. Please check your paper very carefully.

Author Response

We thank the Reviewer again for carefully checking the manuscript and for constructive comments. 

Q1. The authors have corrected the paper according to my comments. However, some minor revisions are still needed. Please change NO to NO thorough the manuscript. NADH- oxidoreductase should be NADH oxidoreductase. Please check your paper very carefully.

A1. We have now checked again and corrected all cases of "NO" and "NADH" that we could find in the manuscript, according to the Reviewer's suggestions.

Reviewer 2 Report

Thank the authors for the corrections in revised manuscript.

However, there were still issues to be corrected sufficiently.

# What was the reason for picking up cerebral ischemia? If authors would like to focus on neurological specialities, the tile of the manuscript should include the words. Indeed, in other organ ischemia such as myocardial ischemia, NO has an important role (PMID: 14962472, 30510628).

# I think issues in “NO and the thiol-disulfide system” were limited within neuron. More specific subtitle is preferable. That is also the case in “4.7. Inhibitors of NOS isoforms and their cytoprotective effect”.

Author Response

We thank again the Reviewer for the specific and valuable comments. We have now modified the manuscript and the answers are below.

Q1. # What was the reason for picking up cerebral ischemia? If authors would like to focus on neurological specialities, the tile of the manuscript should include the words. Indeed, in other organ ischemia such as myocardial ischemia, NO has an important role (PMID: 14962472, 30510628).

A1. In the manuscript, we now used several examples of organs and systems to illustrate the phenomenon, in addition to cerebral ischemia. In particular, we presented the results of studies of changes in the nitroxidergic system in liver, kidney, vascular endothelium, brain, and heart. We also reviewed experimental data on changes in the NO system, from protective to destructive activity in brain and myocardial ischaemia. 

Q2. # I think issues in “NO and the thiol-disulfide system” were limited within neuron. More specific subtitle is preferable. That is also the case in “4.7. Inhibitors of NOS isoforms and their cytoprotective effect”.

A2. The subtitle is now revised and updated according to the Reviewer's comments